# Simultaneous Quantification of Aflatoxin B_1_, T-2 Toxin, Ochratoxin A and Deoxynivalenol in Dried Seafood Products by LC-MS/MS

**DOI:** 10.3390/toxins12080488

**Published:** 2020-07-30

**Authors:** Yijia Deng, Yaling Wang, Qi Deng, Lijun Sun, Rundong Wang, Xiaobo Wang, Jianmeng Liao, Ravi Gooneratne

**Affiliations:** 1Guangdong Provincial Key Laboratory of Aquatic Product Processing and Safety, Guangdong Provincial Engineering Laboratory for Marine Biological Products, Guangdong Provincial Engineering Technology Research Center of Marine Food, Key Laboratory of Advanced Processing of Aquatic Product of Guangdong Higher Education Institution, College of Food Science and Technology, Guangdong Ocean University, Zhanjiang 524088, China; ikea7713@163.com (Y.D.); wangyaling@stu.gdou.edu.cn (Y.W.); wangxiaobo@126.com (X.W.); 2School of Chemistry and Chemical Engineering, Lingnan Normal University, Zhanjiang 524048, China; wangrundong@lingnan.edu.cn; 3Zhanjiang Institute for Food and Drug Control, Zhanjiang 524022, China; zjifdc@163.com; 4Department of Wine, Food and Molecular Biosciences, Faculty of Agriculture and Life Sciences, Lincoln University, P.O. Box 85084, Lincoln 7647, New Zealand; Ravi.Gooneratne@lincoln.ac.nz

**Keywords:** mycotoxins, dried seafood, liquid chromatography-tandem mass spectrometry, method validation, ultrasound-assisted extraction

## Abstract

Mycotoxins are secondary metabolites produced by fungi. These contaminate dried seafoods during processing and storage and represent a potential health hazard for consumers. A sensitive, selective and accurate liquid chromatography/tandem mass spectrometry (LC-MS/MS) method was established for simultaneous quantification of four common mycotoxins (aflatoxin B_1_ (AFB_1_), T-2 toxin (T-2), ochratoxin A (OTA) and deoxynivalenol (DON)) in dried shrimp, dried fish and dried mussel products. Mycotoxins were extracted from dried seafood samples by acetonitrile/water (85/15, *v/v*), subjected to ultrasound for 60 min at 20 °C and cleaned up by defatting with n-hexane. The sample matrix affected the linearity of detection (*R*^2^ ≥ 0.9974). The limit of detection (LOD) and limit of quantification (LOQ) in dried seafood products varied from 0.1 to 2.0 µg·kg^−1^ and 0.3 to 5.0 µg·kg^−1^, respectively. The method was validated by spiking samples with specific mycotoxin levels, and the recoveries, intra-relative standard deviation (RSDs) and inter-RSDs ranged between 72.2–98.4%, 2.8–10.6%, and 5.5–15.4%, respectively. This method was used to analyze 40 dried seafood products purchased from the Zhanjiang seafood market. Results of this product sampling showed that while no DON was detected, AFB_1_, T-2 and OTA were detected in 30.8%, 17.5% and 33.3% of the samples, respectively. AFB_1_, T-2 and OTA concentrations varied at 0.58–0.89, 0.55–1.34 and 0.36–1.51 µg·kg^−1^, respectively. Relatively high frequency of contamination and the presence of AFB_1_, OTA and T-2 residues indicate the need to monitor mycotoxins in dried seafood products.

## 1. Introduction

Seafoods are popular products, rich in proteins, vitamins and minerals, that are consumed by many people, especially those living in coastal regions. In hot, humid environments, fresh seafoods quickly spoil and drying and salting are effective ways to keep seafood edible for longer periods. These storage methods are aimed at reducing water activity to inhibit growth of spoilage microorganisms and inactivate autolytic enzymes [1,2]. Special desirable flavors are produced during the curing and drying process [3,4], making salted and dried seafood products popular for many consumers. However, in many Asian and African countries where fishermen operate on a small scale, for convenience, fresh seafood is dried/smoked in streets or open balconies of houses. This can lead to fungal contamination because of the unhygienic drying conditions. In recent years, several studies have reported the serious fungal load in dried seafood products, especially in dried fish [5,6]. The common fungal genera isolated from contaminate dried fish are *Aspergillus* sp., *Penicillium* sp., *Rhizopus* sp., and *Fusarium* sp. [7,8,9]. Zhanjiang is located in the southern part of China. It is usually hot and humid which is ideal for fungal growth. Our research team previously investigated the fungal diversity of dried seafood in the Zhanjiang market, and found the most common contaminated fungi as *Aspergillus* sp. (especially *A. flavus*), *Fusarium* sp. *Penicillium* sp. and *Trichoderma* sp. (unpublished observation). Residues of mycotoxins produced by these fungi may be found in dried seafood products, which can cause serious food safety issues.

Mycotoxins are naturally toxic secondary metabolites produced by various molds or filamentous fungi species [10]. These can bioaccumulate in the food chain and cannot not usually degraded. Although approximately 350 mycotoxins are currently known, aflatoxins (AFs), ochratoxin A (OTA), deoxynivalenol (DON), T-2 toxin (T-2), zearalenone and fumonisins are the most toxic and have been assigned legal maximum residue limits, a trading standard for food safety [11]. In recent years, reports of mycotoxin contamination in dried seafood products have increased [12,13]. Fagbohun et al. [14] reported 2.73–4.03 µg·kg^−1^ AFB_1_ and 2.01–3.53 µg·kg^−1^ AFG_1_ in 50 smoked-dried fish samples purchased from a major market in Ado-Ekiti, Nigeria. Trace amounts of OTA (1.9 µg·kg^−1^) were reported in dried fish samples from Shanghai [15]. Dietary exposure to AFB_1_ may induce abdominal pain, vomiting, hepatitis, and hepatocellular carcinoma [16] in consumers. OTA can cause renal toxicity including genito-urinary malignancy [17]. Therefore, to improve food safety, it is important to monitor mycotoxins in dried seafood products sold at markets.

Various methods for mycotoxin analysis have been established, such as the chemiluminescent enzyme-linked immunosorbent assay (CL-ELISA) for OTA determination [18], the competitive surface-enhanced Raman scattering (SERS) immunoassay for zearalenone [19] and ultra-performance liquid chromatography (UPLC) with photodiode array for HT-2 and T-2 toxins [20]. In seafood products, AFB_1_ is commonly detected by high performance liquid chromatography (HPLC) [21]. However, a single mycotoxin determination cannot effectively evaluate the overall contamination level in food products.

Dried seafood products are high in protein and low in moisture content and thus are significantly different in matrix to fresh seafood. Currently, liquid chromatography tandem mass spectrometry (LC-MS/MS) is the technique of choice for multiple mycotoxin determination in food. Optimized LC-MS/MS conditions have, for example, been established for simultaneous analyses of the common mycotoxins in coffee beverages [22], animal-derived food [23], breakfast cereals and baby food [24]. An LC-MS/MS method for simultaneous determination of multiple mycotoxins in fresh fish and dried seafood has been reported [15], but it is not that sensitive because of the significant differences in matrix composition between fresh fish and dried seafoods. Ultrasound-assisted extraction (UAE) is an efficient environmentally friendly technique to extract organic contaminants. This method has a lower solvent volume requirement and shorter extraction time compared with classical extraction procedures [25,26,27]. Currently, it is widely used to extract persistent organic pollutants [28], fluoroquinolones [29] and polychlorinated biphenyls [30] in animal tissues. The selection of optimal solvent, temperature, and extraction time in UAE has significantly improved the efficiency of mycotoxin extraction in animal-derived food [23,31].

The objective of this study was to develop a sensitive and accurate LC-MS/MS method for simultaneous determination of four frequently occurring mycotoxins (AFB_1_, T-2, OTA and DON) in dried seafood products.

## 2. Results and Discussion

### 2.1. Optimization of LC-MS/MS Conditions

To obtain good resolution and high sensitivity with LC-MS/MS, mobile phase selection should be aimed at achieving high ionization efficiency of target compounds. In this study, methanol (phase A) and water (phase B) were chosen to test the separation and ionization efficiency of the four target mycotoxins. Results showed good responses for AFB_1_, T-2, and OTA, with peak shapes detected, but not for DON because it did not ionize. Organic acid and ammonium salts added to the mobile phase can significantly improve analyte ionization and increase chromatographic separation efficiency [29]. Therefore, phase B was changed to also contain 5 mmol·L^−1^ ammonium acetate, and this resulted in a DON peak response, but decreased the [M + H]^+^ response. Formic acid (0.1%) added to the mobile phase can greatly improve the [M + H]^+^ responses [23]. Therefore, for better sensitivity and to improve the response from the four mycotoxins, methanol (A) and water containing 5 mmol·L^−1^ ammonium acetate and 0.1% formic acid (B) were selected as the mobile phase. Following optimization of the gradient elution program, the chromatographs thus showed satisfactory peak shapes and good separation efficiency for the four mycotoxins.

To optimize the MS/MS parameters, 1 mg·L^−1^ of each of the mycotoxin standard solutions were separately injected into the MS. Precursor ions were identified in the selected reaction-monitoring (SRM) mode with sharp peaks. Results showed that AFB_1_, OTA and DON formed [M + H]^+^ adducts while T-2 formed [M + NH4]^+^ adducts. For each of the four mycotoxins, two product ions of each precursor ion (Table 1) were selected for optimal sensitivity and selectivity. The SRM chromatograms of the standards of the four mycotoxins are shown in Figure 1.

### 2.2. Optimization of the UAE Extraction Procedure

As well as having a lower moisture content than fresh seafood, the dried products are complex in composition and contain variable fats, proteins, carbohydrates and inorganic salts. Therefore, it is not easy to extract mycotoxins from dried seafood matrices with low interference and high recovery. UAE is an effective extraction method widely used in determining mycotoxin residues in food products [32,33,34]. However, due to the differences in the matrices of the seafoods tested, it was necessary to optimize this extraction method for our purpose.

#### 2.2.1. Extraction Solvents

Generally, due to the good dissolvability of mycotoxins in methanol and acetonitrile, these two organic phases are widely used for simultaneous examination of different mycotoxins in food [26,27]. Four extraction solvents (acetonitrile/water/acetic acid (79/20/1, *v*/*v*/*v*), acetonitrile/water (80/20, *v*/*v*), methanol/water (80/20) and ethyl acetate) [35,36] were selected to compare the extraction efficiency of the four mycotoxins. Among them, acetonitrile/water (80/20, *v*/*v*) showed the best extraction efficiency for all four mycotoxins, ranging from 87.3% to 97.7% (Figure 2A). DON could not be extracted well by the other three solvents (56.3–73.2%). These results agree with the findings of She et al. [33] and Chen et al. [23].

#### 2.2.2. Solvent Extraction Procedure

Different ratios of acetonitrile/water (*v*/*v*, 75/25, 80/20, 85/15, 90/10, 95/5) were tested to evaluate efficiency of extraction of the four mycotoxins. As shown in Figure 2B, acetonitrile/water (*v*/*v*, 75/25) was not suitable for AFB_1_ extraction. The recovery of DON declined gradually with increasing in acetonitrile. The extraction of the four mycotoxins was highest (95.7% to 98.6%) with 85/15 acetonitrile/water (*v*/*v*). Extraction temperature (0, 10, 20, 30, 40 °C) were tested to select the most efficient temperature for recovery of the four mycotoxins. Results showed that recovery of the four mycotoxins gradually increased from 0 °C to 20 °C, varying from 80% to 100%. Increasing the temperature (to 30–40 °C) did not significantly improve recovery (Figure 2C) and there was also concern about the co-extraction of a larger fraction of the soluble organics. Next, different extraction times (20, 40, 60, 80, 100 min) were tested. It was found that recovery of the four mycotoxins significantly increased over time with a recovery of up to 80% at 60 min, which was similar to that observed at 80 and 100 min (Figure 2D). Therefore, 60 min was selected. As shown in Figure 2E, DON could not be fully extracted at pH 3 (72.5%), and the recovery of OTA was low (76.4%) at pH 9. The recoveries of all four mycotoxins were high (>80%) at pH 7 but highest (87.1–97.9%) when extracted without pH adjustment. Hence, the pH of the extraction solution was not adjusted from the original (pH 6.3).

### 2.3. Optimization of the Clean-Up Procedure

Before LC-MS/MS detection, a clean-up procedure is usually employed following extraction to minimize matrix interferences and to concentrate the mycotoxins [37]. High recovery of the four mycotoxins, ranging from 87.9 to 92.8% and 87.8 to 100.4%, were detected following n-hexane and immunoaffinity purification procedures, respectively. However, DON and OTA could not be extracted well with the alumina N-neutral column purification and OTA recovery was lowest with the SiO_2_-solid phase extraction column (Figure 2F). Although immunoaffinity column clean-up was the most effective method and has been the most widely used for mycotoxins [38], it is time-consuming and uneconomical. To establish a quick, cheap, and effective method that could be widely used, the n-hexane-defatting purification method was selected.

### 2.4. Linearity

Linearity (with coefficients > 0.9989) of the different concentrations of the four mycotoxin standard solutions are shown in Table 2.

### 2.5. Matrix Effects

Linearity of calibration curves for the four mycotoxins in three seafood matrices, with seven concentration sequences, were evaluated (Table 3). *R*^2^ of matrix-matched calibration regression of all dried seafood samples ranged from 0.9974 to 0.9999. A good linear relationship was also achieved for AFB_1_, T-2, OTA and DON in all samples tested. The LOD ranged from 0.1 to 2.0 μg·kg^−1^ and the LOQ from 0.3 to 5.0 μg·kg^−1^ in the diverse samples. There were no obvious matrix effects in the dried shrimp and dried fish products. However, a significant matrix effect was detected in the dried mussel samples. The LOD of the four mycotoxins in dried mussel matrix was 0.3–1 μg·kg^−1^, higher than the LOD of the standard solutions (0.1–1 μg·kg^−1^). In general, an signal suppression/enhancement (SSE)(%) of 80–120% is acceptable. Results showed that no significant matrix effects were observed for the four mycotoxins in dried shrimp (93.4–100.6%) and dried fish products (90.1–103.8%). A slight matrix effect was detected in dried mussel (87.6–94.5%) but still higher than the minimum acceptable level of 80%. This may be because mussel is nutritionally more dense, with higher concentrations of carbohydrates, minerals, fatty acids [39] and also trace elements such as copper (Cu), manganese (Mn), selenium (Se) and zinc (Zn), than shrimp and fish [40]. The presence of these substances increases the matrix effects, which can affect the method accuracy.

### 2.6. Method Validation

The recoveries, intra- and inter-day precision for low, intermediate, and high concentrations of the four mycotoxins are shown in Table 4. The recovery ranges for the four mycotoxins were 72.2–98.4% from dried shrimp, 77.3–96.5% from dried fish and 74.1–97.6% from dried mussel. Generally, recovery for mycotoxins in the range of 70–120% is regarded as acceptable [15]. The intra-relative standard deviation (RSD) and inter-RSD values were in the range of 2.8–10.6% and 5.5–15.4%, respectively. Therefore, based on all validation parameters and test results, the method could be considered as suitable for simultaneous AFB_1_, T-2, OTA and DON detection in dried seafood products.

### 2.7. Real Sample Analysis

The validated LC-MS/MS method was applied to analyze the mycotoxin residues in 40 dried seafood samples purchased from the Zhanjiang market. Results showed that AFB_1_, T-2 and OTA commonly exist in dried seafood including shrimp, mussel, scallop, octopus and fish products. DON was not detected in any of the samples tested. AFB_1_ and OTA were the most frequent contaminants in the dried seafood products with an incidence of 30.8% and 33.3%, respectively > T-2 (17.5%). The concentration of AFB_1_ in all seafood samples where it was detected ranged from 0.58 to 0.87 μg·kg^−1^ and OTA from 0.36 to 1.51 μg·kg^−1^. T-2 residue in all samples where it was detected ranged from 0.55 to 1.34 μg·kg^−1^ (Table 5).

There are three possible routes by which mycotoxins contaminate dried seafood products: (i) during the curing or drying processes when airborne microorganisms and/or their spores fall on the surface of the drying seafood, with subsequent fungal growth and mycotoxin production; (ii) improper storage such as exposure to high temperatures and/or high humidity; (iii) fungi transferred from consumers when they select the dried seafood products at the market by hand. A variety of filamentous molds such as *Aspergillus* and *Penicillium* spp. grow competitively on dried seafood products and result in subsequent mycotoxin production. Mycotoxin contamination of dried seafood products appears to be widespread. For example, 1.9 μg·kg^−1^ OTA was detected in a dried fish sample from Shanghai [15] and aflatoxin residues in dried fish products have been reported in Nigeria (1.05–25.00 μg·kg^−1^) [41], Zambia (average 23 μg·kg^−1^) [5] and India (1.3–3.84 μg·kg^−1^) [12], all of which are tropical countries. Zhanjiang is recognized as having a tropical climate, with high temperatures and humidity during most of the year, and mold contamination occurs frequently. To our knowledge, this is the first time that mycotoxin residues have been demonstrated in dried seafood products in this region. Although the mycotoxin residues in the dried seafood samples were at low concentration, the occurrence frequency was high. Therefore, as a food safety measure, it is important to monitor mycotoxin residues in dried seafood products in Zhanjiang and other regions with hot and humid climatic conditions.

## 3. Conclusions

A reliable, sensitive, cost-effective, fast and efficient analytical LC-MS/MS method for simultaneous determination of AFB_1_, T-2, OTA and DON in three dried seafood product types was established. Different parameters that influence the extraction efficiency and detection sensitivity of the method were studied. Mycotoxins in dried seafood samples were subjected to ultrasound-assisted acetonitrile/water (85/15) extraction for 60 min at 20 °C with an n-hexane-defatting clean-up procedure. The method showed good linearity, precision, and recovery. The high frequency of occurrence of mycotoxins in our samples of commercial dried seafood indicates the necessity to monitor mycotoxins in such products.

## 4. Materials and Method

### 4.1. Reagents and Solutions

HPLC-grade acetonitrile, ammonium acetate, formic acid and methanol were purchased from Sigma-Aldrich (St. Louis, MO, USA). Milli-Q water was purified using an Arium 611 VF (Sartorius, Germany). N-hexane, sodium hydroxide, hydrochloric acid, ethyl acetate and sodium chloride were of analytical grade and purchased from Qiyun (Guangdong, China). The standards (≥98% purity) of AFB_1_, T-2, OTA and DON were purchased from Enzo Life Science (Farmingdale, NY, USA). Stock solutions of each of the four mycotoxins were prepared by dissolving 1 mg of the respective mycotoxin standard in 10 mL of acetonitrile (0.1 mg·mL^−1^), and then were stored at −20 °C. Working standard solutions (AFB_1_, OTA (100 μg·L^−1^), T-2 (200 μg·L^−1^) and DON (500 μg·L^−1^)) were prepared by diluting the stock solutions with methanol/water (30/70, *v*/*v*) mixed with 5 mmol·L^−1^ ammonium acetate, before being stored at 4 °C in preparation for optimization analyses.

### 4.2. Sample Preparation

Forty dried seafood products of three types, dried fish (*Lutjanus sanguineus*), dried shrimp (*Litopenaeus vannamei*) and dried mussel (*Mytilus edulis*) (~500 g per seafood product), were purchased from the seafood market in Zhanjiang, China, to determine mycotoxin contamination in dried seafood. The dried fish were cut open and the bones were removed. The muscle of dried shrimps was extracted from the shells. The edible part of dried mussel was collected. The edible parts of each dried seafood product (200 g muscle) were dried at 50 °C for 12 h, milled for 2 min using a food grinder (LX-20B, Langxin, China) to obtain the sample powders, and stored at −20 °C until further analysis. Mycotoxin detection was undertaken three times per sample and each analytical parameter was optimized one at a time using 2 g for each analytical procedure.

### 4.3. Optimization of Extraction Solvent

To determine the optimal solvent for extraction of multiple mycotoxins, 2 g sample powder of each of the three seafood products were weighed and added to 40 μL mixture of mycotoxin standard solution containing 20 μg·kg^−1^ of each of AFB_1_, T-2, OTA and DON mycotoxins. The mycotoxins were extracted with either 10 mL of acetonitrile/water/acetic acid (79/20/1, *v*/*v*/*v*), acetonitrile/water (80/20), methanol/water (80/20) or ethyl acetate, respectively. The solution mixture was subjected to UAE (PS-30 A, power: 180 W, frequency: 40 kHz, Ruimi Instruments Co., Changzhou, China) at 40 °C for 20 min. After UAE, samples were centrifuged at 4500× *g* for 10 min. This extraction process was repeated twice more, and the pooled supernatants were stored at −20 °C until LC–MS/MS analysis. Each sample type was analyzed five times.

### 4.4. Optimization of Extraction Solvent Ratio

Based on the optimal extraction solvent selected by the method in 4.3, the ratio of sample/solution mixture to extraction solvent (*v*/*v*) was adjusted to 75/25, 80/20, 85/15, 90/10, and 95/5 to determine the ratio that gave the highest recovery of the four mycotoxins.

### 4.5. Optimization of UAE Extraction Condition

After the extraction solvent ratio was selected, the extraction temperature (0, 10, 20, 30 and 40 °C), extraction time (20, 40, 60, 80 and 100 min) and extraction pH (3, 7 and 9; adjusted using 1.5 mol·L^−1^ NaOH or 1 mol·L^−1^ HCl) to determine the optimal UAE extraction conditions for maximal recovery of four mycotoxins were determined.

### 4.6. Optimization of Clean-Up Procedure

#### 4.6.1. N-Hexane-Degreasing Purification

The 5 mL of the extracted supernatant was evaporated to dryness in a nitrogen (N_2_) stream at 50 °C (DC12H, ANPFL Scientific Co., Shanghai, China), then re-dissolved in 1 mL of methanol/water (30/70, *v*/*v*) with 0.1% formic acid, and 1 mL of n-hexane was added for defatting. The mixture was vortexed for 2 min then allowed to sit for 20 min at room temperature. The lower phase was extracted and filtered through a 0.22 μm MillexGV membrane filter (Millipore, Molsheim, France) before analysis.

#### 4.6.2. Immunoaffinity Column Purification

The 5 mL of extracted supernatant was loaded onto an immunoaffinity column (Pribolab, Biotech Co., Qingdao, China) at a rate of 2 mL/min until 2–3 mL of air had passed through the column. Then, the column was washed by double-distilled water at a rate of 1 drop/s until 2–3 mL of air had passed through the column. Next, 3 mL of anhydrous ethanol was used to elute the analytes at a rate of 1 drop/s. All eluants were collected in a 10 mL nitrogen blowpipe to dry though N_2_ at 50 °C. The residue was re-dissolved in 1 mL of methanol/ammonium acetate (30/70, *v*/*v*) and the solution filtered through a 0.22 μm filter before analysis.

#### 4.6.3. Alumina N-neutral Column Purification

An alumina N-neutral column (AISIMO, BCTC Co., Shanghai, China) was cleaned by 10 mL of acetonitrile, then 5 mL of the extracted supernatant was passed through the column at a rate of 2 mL/min. A 10 mL nitrogen blowpipe was used to collect the eluate. Next, 6 mL of acetonitrile was used to clean up the elution and the eluate transferred to a tube and evaporated by N_2_ at 50 °C. The residue was re-dissolved in 1 mL of methanol/ammonium acetate (30/70, *v*/*v*) and the solution filtered through a 0.22 μm filter before analysis.

#### 4.6.4. SiO_2_-Solid Phase Extraction Column Purification

Before purification, 6 mL of methanol was used to clean the column. Next, 5 mL of the extracted supernatant was passed through a SiO_2_-solid phase extraction column (Simo Aldrich, Germany) at a rate of 2 mL/min. Then, 6 mL of purified ultrapure water followed by 6 mL of acetonitrile were used to clean up the elution and the eluate was transferred into a test tube and evaporated with N_2_ at 50 °C. The residue was re-dissolved in 1 mL of methanol/ammonium acetate (30/70, *v*/*v*) and the solution filtered through a 0.22 μm filter before analysis.

### 4.7. LC-MS/MS Operating Conditions

Mycotoxin analysis was performed on a Thermo Scientific Surveyor HPLC system (Thermo Scientific, CA, USA) that comprised a Surveyor MS Pump Plus, an on-line degasser, and a Surveyor Autosampler Plus coupled with a Thermo TSQ Quantum Access tandem mass spectrometer equipped with an electrospray ionization (ESI) source (Thermo Scientific, San Jose, CA, USA). The separation was performed at 35 °C using a Hypersil GOLD column (100 mm × 2.1 mm, 5 μm) (Thermo Fisher Scientific, San Jose, CA, USA) with a flow rate of 0.25 mL per min. The injection volume was 5 μL. The analytical time in the MS/MS system was 10 min. The mobile phase consisted of methanol (A) and water containing 5 mM of ammonium acetate and 0.1% formic acid (B). The gradient elution program was as follows: 0 min, 30% A; 3.0 min, 90% A; 5 min, 90% A; from 5.1 min to 8 min, 30% A; at the end of the process for 2 more min for re-equilibration.

MS/MS detection was carried out using a triple quadruple mass spectrometer, coupled with an electrospray ionization source operated in positive mode (ESI+) and negative mode (ESI-). Quantitation was performed in the multiple reaction monitoring (MRM) mode with the positive mode (ESI+) and negative mode (ESI-) scanned simultaneously. The working conditions for MS/MS were as follows: spray voltage: 4500 V; auxiliary gas pressure: 15 au; sheath gas pressure: 35 au; capillary temperature: 350 °C; collision energy: 1.5 eV; tube lens offset: 118 V; and collision pressure: 1.5 mTorr. Product ion scan and the selected reaction-monitoring (SRM) modes were used for mass spectrometer operation. For each of the mycotoxins, two selected product ions of the precursor ions were monitored in the SRM mode and one of the product ions was used for quantification (AFB_1_, 241.0; T-2, 185.1; OTA, 358.1; DON, 249.1).

### 4.8. Linearity

Linearity of the method was evaluated by making seven concentrations of each mycotoxin in a mixed standard solutions (T-2: 0.5, 1, 10, 20, 40, 100, 200 μg·kg^−1^; AFB_1_, OTA: 0.5, 1, 5, 10, 20, 50, 100 μg·kg^−1^; DON: 1, 5, 10, 50, 100, 200, 500 μg·kg^−1^) as described in Section 2.1. Each mycotoxin standard concentration was analyzed in triplicate to derive the calibration curves. The peak area was used for quantification of the sample mycotoxins.

### 4.9. Matrix Effects and Method Validation

The effects of matrix (fish/shrimp/mussel) were evaluated for each mycotoxin by comparing the calibration curves of “pure” stock solutions of mycotoxin standards diluted with “blank” matrices (seafood powder without added mycotoxin standard extracted by acetonitrile/water, 80/20, *v*/*v*, containing 5 mmol L^−1^ ammonium acetate and 0.1% formic acid) alone and the calibration curves of the final extracted solution spiked with mycotoxins. The 2 g blank samples were pretreated by UAE extraction (extracted by acetonitrile/water (85/15, *v*/*v*), subjected to ultrasound for 60 min at 20 °C) and the final extracted solution was obtained. Then, it was added with seven concentrations of target mycotoxin, resulting in matrix-matched solutions. Each solution was analyzed in triplicate. Each concentration of standard mycotoxin solution was compared with the slopes of matrix-matched calibration solutions to assess the matrix effects [28].

Linearity, sensitivity (limits of detection (LOD), limits of quantitation (LOQ)), recovery and precision (RSDr (intra-day precision) and RSD_R_ (inter-day precision)) were used to validate our optimized method for simultaneously quantitative analysis of AFB_1_, T-2, OTA and DON in dried seafoods. Calibration curves were constructed by plotting the response values of the concentration of each mycotoxin in the different matrices. The sensitivity of the method was evaluated by determining the LOD and LOQ using stock standard solutions diluted with the blank (unspiked samples of each seafood matrix) dried seafood matrices. LOD refers to the concentration of the four mycotoxins in each matrix that provided a signal-to-noise ratio (S/N) of 3/1, while LOQ is the concentration with S/N of 10/1. Intra-day precision (RSD_r_) and inter-day precision (RSD_R_) were evaluated using blank samples (the original three seafood matrix solutions) and comparing their results with those of the same solutions spiked with low, medium and high levels of each mycotoxin as a mixture (AFB_1_, OTA: 1, 50, 100 μg·kg^−1^; T-2: 1, 100, 200 μg·kg^−1^; DON: 2, 200, 400 μg·kg^−1^). Powdered dried seafood sample solutions were prepared in quintuplicate to calculate the RSD_r_ evaluation. Sample solutions spiked with the same concentration of mycotoxin standard were pretreated in quintuplicate for 5 consecutive days for RSD_R_ evaluation. Extraction recoveries of each mycotoxin in the different matrices were evaluated as follows:Recovery (%) = observed concentration/spiked concentration (theoretical value) × 100

### 4.10. Statistical Analysis

All experiments were performed in parallel five times and all data from these studies are expressed as mean ± S.E.M. Statistical computations were performed with SPSS 19.0 (SPSS Inc., Chicago, IL, USA) and Origin 8.5 (Origin Lab Inc., Northampton, MA, USA) software.

## Figures and Tables

**Figure 1 toxins-12-00488-f001:**
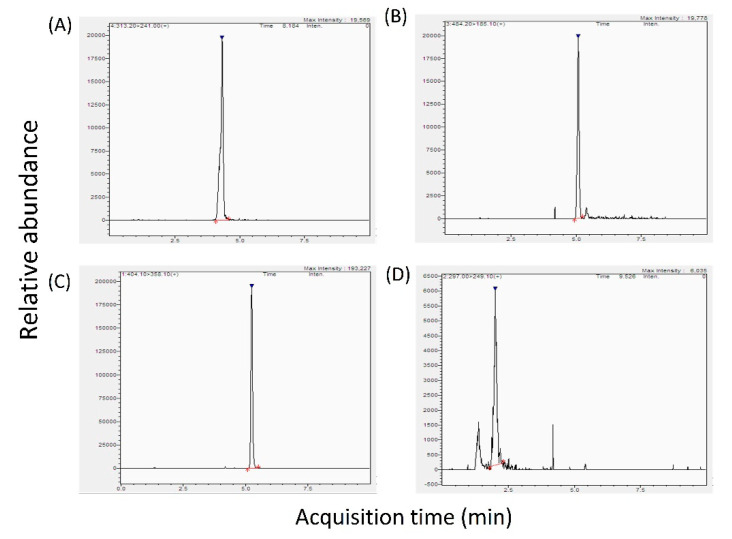
LC-MS/MS chromatograms of the four mycotoxins: (**A**) aflatoxin B_1_ (AFB_1_), (**B**) T-2 toxin, (**C**) ochratoxin A (OTA), and (**D**) deoxynivalenol (DON) in dried seafood products spiked with 1 mg·L^−^^1^ of each of the standards.

**Figure 2 toxins-12-00488-f002:**
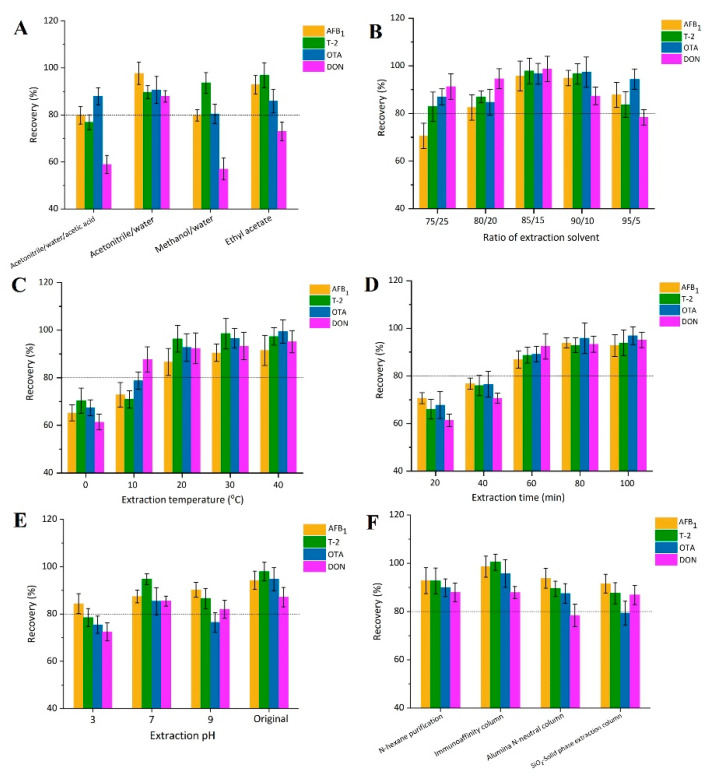
Recovery efficiency of (**A**) different extraction solvents, (**B**) ratio of extraction solvents, (**C**) extraction temperature, (**D**) extraction time, (**E**) extraction pH, and (**F**) clean-up columns for aflatoxin B_1_ (AFB_1_), T-2 toxin (T-2), ochratoxin A (OTA) and deoxynivalenol (DON) in three types of dried seafood product spiked with a 20 μg·L^−1^ mixture of the mycotoxin standards (*n* = 5).

**Table 1 toxins-12-00488-t001:** Optimum precursor and product analyte ions with the respective collision energy values (CE (eV)) for MS/MS.

Toxin	Precursor Ion (*m/z*)	Product Ions (*m/z*)	CE (eV)	Retention Time (min)
AFB_1_	313.2 [M + H]^+^	213.0	44	4.3
		241.0 ^a^	36	
T-2	484.3 [M + NH4]^+^	185.1 ^a^	27	5.1
		215.1	25	
OTA	404.0 [M + H]^+^	238.1	27	5.3
		358.1 ^a^	15	
DON	297.3 [M + H]^+^	203.1	16	2.0
		249.1 ^a^	12	

^a^ Quantitative ion; AFB_1_: aflatoxin B_1_, T-2: T-2 toxin, OTA: ochratoxin A, DON: deoxynivalenol.

**Table 2 toxins-12-00488-t002:** Parameters of the calibration curves for the four mycotoxin standards.

Toxin	Linear Range (μg·kg^−1^)	Standard Calibration	*R* ^2^	Slope	LOD (μg·kg^−1^)	LOQ (μg·kg^−1^)
AFB_1_	0.5–100	*y* = 1014.5*x* − 321.19	0.9989	1014.5	0.1	0.3
T-2	0.5–200	*y* = 885.44*x* + 278.19	0.9994	885.4	0.1	0.5
OTA	0.5–100	*Y* = 7451.49*x* − 952.7	0.9999	7451.5	0.1	0.5
DON	1–500	*Y* = 355.52*x* − 339.42	0.9990	355.5	1.0	3.0

LOD: limit of detection, LOQ: limit of quantification, AFB_1_: aflatoxin B_1_, T-2: T-2 toxin, OTA: ochratoxin A, DON: deoxynivalenol.

**Table 3 toxins-12-00488-t003:** The matrix-matched calibration curves and sensitivities of the four mycotoxins in dried seafood.

Matrix	Mycotoxin	Linear Range (μg·kg^−1^)	Standard Calibration ^a^	*R* ^2^	Slope	LOD (μg·kg^−1^)	LOQ (μg·kg^−1^)	SSE (%)
Dried shrimp	AFB_1_	0.5–100	*y* = 1047.21*x* + 289.35	0.9999	1047.2	0.1	0.5	100.6
	T-2	0.5–200	*y* = 653.31*x* + 357.65	0.9999	653.3	0.2	0.5	95.7
	OTA	0.5–100	*y* = 7302.98*x* + 523.42	0.9979	7303.0	0.1	0.3	95.2
	DON	1–500	*y* = 241.77*x* + 921.42	0.9987	241.8	2.0	3.0	93.4
Dried fish	AFB_1_	0.5–100	*y* = 1223.01*x* + 542.5	0.9989	1223	0.1	0.5	103.8
	T-2	0.5–200	*y* = 783.41*x* + 892.97	0.9989	783.4	0.5	1.0	95.4
	OTA	0.5–100	*y* = 6862.12*x* + 399.11	0.9999	6862.1	0.3	1.0	92.3
	DON	1–500	*y* = 268.23*x* + 223.42	0.9999	268.2	1.0	5.0	90.1
Dried mussel	AFB_1_	0.5–100	*y* = 1008*x* + 509.12	0.9989	1008	0.5	1.0	94.5
	T-2	0.5–200	*y* = 589.3*x* + 305.31	0.9979	889.3	0.5	1.0	86.6
	OTA	0.5–100	*y* = 5972.19*x* + 423.1	0.9985	5972.2	0.3	1.0	89.5
	DON	1–500	*y* = 233.98*x* + 692.44	0.9974	234	1.0	5.0	87.6

^a^ Calibration curves for the mycotoxins in the seafood samples before being spiked with additional mycotoxins at seven concentrations. (T-2: 0.5, 1, 10, 20, 40, 100, 200 μg·kg^−1^; AFB_1_, OTA: 0.5, 1, 5, 10, 20, 50, 100 μg·kg^−1^; DON: 1, 5, 10, 50, 100, 200, 500 μg·kg^−1^); LOD: limit of detection, LOQ: limit of quantification, AFB_1_: aflatoxin B_1_, T-2: T-2 toxin, OTA: ochratoxin A, DON: deoxynivalenol.

**Table 4 toxins-12-00488-t004:** Recovery, intra-day precision and inter-day precision of the four target mycotoxins in three dried seafood products (*n* = 5).

Mycotoxin	Spiked Concentration (μg·kg^−1^)	Dried Shrimp			Dried Fish			Dried Mussel		
		Recovery (%)	RSD_r_	RSD_R_	Recovery (%)	RSD_r_	RSD_R_	Recovery (%)	RSD_r_	RSD_R_
AFB_1_	1	94.31 ± 4.03	5.6	12.1	96.52 ± 2.63	5.2	15.4	91.53 ± 7.59	6.9	8.9
	50	92.64 ± 3.47	5.8	7.4	89.79 ± 4.60	4.7	7.8	95.04 ± 4.85	6.4	9.8
	100	85.73 ± 8.72	6.2	7.8	92.43 ± 7.68	8.9	8.4	83.26 ± 3.27	4.8	5.7
T-2	1	78.08 ± 6.15	3.8	14.5	89.32 ± 2.67	6.5	6.9	90.47 ± 7.59	5.5	9.0
	100	95.16 ± 9.71	4.4	8.9	85.38 ± 4.26	5.2	12.0	95.04 ± 4.85	2.8	11.9
	200	98.38 ± 7.72	4.0	10.3	95.84 ± 3.91	7.9	7.7	83.26 ± 3.27	3.5	15.1
OTA	1	79.90 ± 3.29	8.6	7.7	87.34 ± 2.43	8.8	10.8	82.84 ± 4.52	10.6	7.4
	50	96.75 ± 5.28	5.2	8.4	94.74 ± 4.92	7.1	9.7	95.40 ± 7.45	4.9	8.0
	100	89.37 ± 5.45	3.4	6.0	90.19 ± 6.75	5.1	5.5	97.59 ± 3.64	7.7	12.7
DON	2	72.24 ± 6.25	9.3	9.6	79.52 ± 6.16	4.9	7.8	74.15 ± 2.71	9.9	10.5
	200	79.65 ± 8.83	7.1	8.6	82.31 ± 4.68	5.4	9.0	85.34 ± 4.85	8.4	10.7
	400	80.57 ± 3.93	5.3	9.6	77.26 ± 5.35	3.3	11.9	83.51 ± 2.39	6.5	11.6

RSD_r_ (%), intra-day precision evaluated by spiking blank samples. RSD_R_ (%), inter-day precision carried out on three different days, evaluated by spiking blank samples. AFB_1_: aflatoxin B_1_, T-2: T-2 toxin, OTA: ochratoxin A, DON: deoxynivalenol.

**Table 5 toxins-12-00488-t005:** Mycotoxin concentrations in samples (*n* = 40) of dried seafood purchased from Zhanjiang fish market.

Seafood Type	Sample Number ^a^	Mycotoxin Residues (μg·kg^−1^)			
		AFB_1_	T-2	OTA	DON
Dried shrimp	1	- ^b^	-	0.41 ± 0.02	-
	2	0.58 ± 0.01	0.65 ± 0.02	0.57 ± 0.03	-
	3	-	0.87 ± 0.04		-
	7	-	-	0.37 ± 0.01	-
Dried mussel	13	0.64 ± 0.06	1.34 ± 0.21		-
Dried scallop	14	-	0.88 ± 0.11	0.36 ± 0.03	-
Dried octopus	15	-	0.55 ± 0.01		-
Dried fish	16	0.58 ± 0.04	-		-
	19	0.72 ± 0.01	1.07 ± 0.16		-
	25	-	-	0.89 ± 0.17	-
	27	0.63 ± 0.03	-	0.43 ± 0.04	-
	28	0.87 ± 0.07	0.61 ± 0.07	0.62 ± 0.07	-
	29	0.59 ± 0.02	-		-
	30	0.83 ± 0.11	-	0.70 ± 0.06	-
	32	0.62 ± 0.05	-	0.51 ± 0.02	-
	34	0.89 ± 0.13	-		-
	35	-	-	0.82 ± 0.15	-
	36	0.78 ± 0.08	-	0.36 ± 0.01	-
	37	0.74 ± 0.12	-		-
	38	-	-	0.73 ± 0.03	-
	39	-	-	1.51 ± 0.14	-

^a^ No mycotoxins were detected in 19/40 samples (dried shrimp (sample 1–7), dried mussel (sample 8–13), dried scallop (sample 14), dried octopus (sample 15), dried fish (sample 16–40)). AFB_1_: aflatoxin B_1_, T-2: T-2 toxin, OTA: ochratoxin A, DON: deoxynivalenol. ^b^ Not detected. - refers to LOD, the limit of detection.

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
