# Peer review of "Simultaneous Quantification of Aflatoxin B1, T-2 Toxin, Ochratoxin A and Deoxynivalenol in Dried Seafood Products by LC-MS/MS"

_toxins, 2020, doi:10.3390/toxins12080488_

Round 1

Reviewer 1 Report

I suggest a major revision.

Main concerns:

Some review papers shall be cited in the introduction.

How the mycotoxin appear in seafood?

Why these toxins were selected?

Were matrix-matched calibration used?

Why the authors did not use isotopically labeled internal standard?

Fig. 1 and 2 have low resolution, poor figures.

Please use µg/kg in each table, convert the µg/L to µg/kg using the dilution/concentration factor.

Section 4.7: stop time? Ion transitions used? Please detail them.

Please detail the slope of both matrix-matched and solvent calibrations. Compare them and add the percentage of matrix effect in all types of matrices.

Reviewer 2 Report

Author must improve the introduction part (include the details about the more seafood product and their contamination, presence of fungal contamination, previous reports)

Figure1 and Figure 2 – Quality was so bad, author must improve the picture resolution.

2.2.1. Extraction solvent

Line no: 112- what about the ethyl acetate ratio?

2.2.2, 2.2.3,2.2.4, 2.2.5- Try to merge the results.

Line no- 167 Dried mussel matrix was 0.3 – 1.0 µg/L-1 But in the Table-3 the range was mentioned 1 to 5.0 µg/L-1 ?

4.5.1, 2,3 – Try to merge the material and methods.

Author must provide the statistical analysis.

Did author check the fungus profile in all seafood products? if so, provide the details as the toxin profile may vary with fungal diversity... It might be correlate the toxin contamination in all seafood products.

Why author did not check the seasonal variation of mycotoxins in all seafood products, after the LC-MS/MS method development?

Round 2

Reviewer 1 Report

The paper can merit publication in this form.